# Neural Poisson Surface Reconstruction: Resolution-Agnostic Shape Reconstruction via Fourier Neural Operators

Hector Andrade-Loarca[*,1,3]     Julius Hege[*,2,3,4]     Daniel Cremers[1,3,4]     Gitta Kutyniok[2,3,4]

[1]Technical University of Munich     [2]Ludwig-Maximilians-Universität in Munich
[3]Munich Center for Machine Learning     [4]relAI – Konrad Zuse School of Excellence in Reliable AI

hector.andrade@tum.de     hege@math.lmu.de     cremers@tum.de     kutyniok@math.lmu.de

## Abstract

*Reconstructing high-quality 3D shapes from sparse or noisy point clouds is a long-standing challenge. Traditional methods struggle with low-quality inputs, while modern learning-based approaches can be computationally demanding or fail to generalize. To bridge this gap, we propose Neural Poisson Surface Reconstruction (nPSR), a novel hybrid method that combines traditional model-based Poisson Surface Reconstruction and learned neural operators for accurate 3D shape reconstruction from oriented point clouds. We solve the classical Poisson reconstruction formulation with Fourier Neural Operators, leveraging their efficiency while learning a robust data-driven prior, significantly enhancing reconstruction quality, especially under sparse or noisy sampling conditions. Importantly, nPSR achieves resolution-agnostic performance, training on low-resolution grids and generalizing effectively to higher resolutions without retraining. Experimental results demonstrate that nPSR outperforms state-of-the-art reconstruction methods when reconstructing from sparse samples and generalizes to unseen datasets. The reconstruction occurs in a single forward pass, allowing for integration into larger differentiable vision pipelines for end-to-end optimization.*

## 1. Introduction

Shape reconstruction, the process of creating accurate three-dimensional (3D) meshes from measurements such as images or point clouds, is fundamental in computer vision, with wide-ranging applications from robotics and virtual reality to autonomous driving. A critical subproblem is reconstructing surfaces from oriented point clouds, which often contain noise and are only sparsely sampled, typically acquired via

---
[*]Equal contribution.

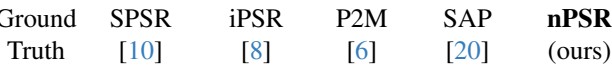

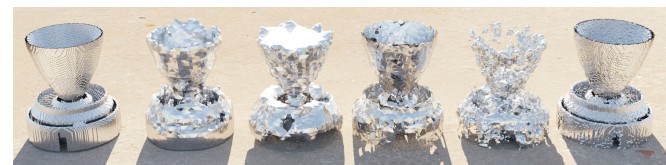

| Ground Truth | SPSR [10] | iPSR [8] | P2M [6] | SAP [20] | **nPSR** (ours) |
|---|---|---|---|---|---|

Figure 1. Whereas baseline methods work well in the high-sampling regime, the proposed resolution-agnostic nPSR method clearly outperforms them in the low-sampling regime. This image shows reconstructions from 3 000 sampled points.

photogrammetry or depth sensors.

Classical algorithms, notably Poisson Surface Reconstruction (PSR) [10] and Ball Pivoting [2], employ handcrafted constraints to reconstruct surfaces reliably. However, these methods struggle significantly in sparse or noisy scenarios due to the limited robustness of handcrafted priors. Recent deep learning approaches like Convolutional Occupancy Networks [19], DeepSDF [18], POCO [3], ALTO [24], and NKSR [9] achieve improved reconstruction quality by learning shape priors from data. However, they frequently fail to generalize beyond the training distribution and underperform in sparse sampling regimes [23].

A key challenge for neural reconstruction approaches is scalability and generalization across resolutions. Standard convolutional neural networks (CNNs) rely on fixed-resolution voxel grids, which limits detail and requires costly retraining at each new resolution. While methods like implicit representations [18, 21] avoid fixed grids, their inference cost grows with output resolution, often making them computationally prohibitive.

To address these limitations, we propose Neural Poisson Surface Reconstruction (nPSR), a novel hybrid framework

that merges the robustness of classical PDE-based PSR with the flexibility of learned neural operators. Specifically, we leverage Fourier Neural Operators (FNOs) [15], chosen for their computational efficiency, PDE-solving capability, and inherent resolution-agnostic properties. This allows training at low resolutions and effective generalization to higher resolutions, supporting efficient one-shot super-resolution.

## 1.1. Main contributions

Our primary contributions are:

- A novel hybrid reconstruction method integrating the classical Poisson surface reconstruction setting with a learned implicit prior, replacing handcrafted boundary conditions and significantly improving fidelity in sparse and noisy scenarios.
- Robust performance in both low- and high-sampling regimes, surpassing state-of-the-art methods under sparse sampling while retaining near-perfect accuracy with dense data and generalizing to unseen datasets.
- The first application of Neural Operators for surface reconstruction, demonstrating their suitability and efficiency in solving 3D reconstruction tasks like the Poisson PDE.
- Efficient resolution-agnostic performance via FNO, allowing training at lower voxel resolutions (e.g., $64\times64\times64$) and high-quality inference at higher resolutions (e.g., $128\times128\times128$) without retraining, addressing scalability issues common in convolutional architectures.
- Fully differentiable reconstruction pipeline enabling integration into larger differentiable vision systems, supporting tasks like shape optimization or inverse modeling.

## 2. Related Work

### 2.1. Poisson Surface Reconstruction

Poisson Surface Reconstruction (PSR, Kazhdan et al. [11]) addresses the problem of reconstructing surfaces from oriented point clouds by formulating it as a PDE-based inverse problem. Given a point cloud with associated normal vectors, PSR reconstructs an indicator function $\chi$ by solving the Poisson equation:

$$\Delta\chi(x) = \nabla \cdot V(x), \quad x \in \Omega, \quad (2.1)$$

subject to homogeneous Dirichlet boundary conditions. Later improvements, such as Screened Poisson Surface Reconstruction (SPSR) [10], have introduced alternative regularizations to enhance reconstruction quality.

More recent developments include differentiable Poisson solvers [20], iterative normal-estimation methods like iPSR [8], and stochastic PDE solutions using Physics-Informed Neural Networks [22]. While highly accurate at dense sampling levels, these approaches tend to produce suboptimal reconstructions at lower sampling densities, struggle to inter-

polate sparse data effectively, and are computationally more demanding.

### 2.2. Fourier Neural Operators

Fourier Neural Operators (FNOs), introduced by Li et al. [15], efficiently approximate solutions of PDEs by learning mappings between infinite-dimensional function spaces. Formally, an FNO iteratively transforms an input function through Fourier-space operations and learned linear transformations. Specifically, given input $a$, the FNO computes the solution $u$ iteratively as:

$$v_{t+1}(x) := \sigma\left(W v_t(x) + \mathcal{F}^{-1}(R \cdot \mathcal{F}(v_t))(x)\right), \quad (2.2)$$

where $\mathcal{F}$ denotes Fourier transform, and $R$ represents learned weights in Fourier domain. The key advantage of FNOs is their computational efficiency ($O(n \log n)$ complexity in resolution), resolution-agnostic properties, and demonstrated universal approximation capabilities [13]. Recent applications include weather modeling [14] and multiphase fluid flow [25]. We leverage these attributes by using FNOs specifically to solve Poisson's equation for 3D shape reconstruction.

### 2.3. Implicit Surface Reconstruction

Implicit neural representations offer another approach to resolution-independent reconstruction by representing surfaces implicitly via level sets of neural networks. DeepSDF [18] pioneered the use of implicit neural models for shape representation. Subsequent works, such as neural kernel fields [26], Neural Dual Contouring (NDC) [5], and GeoUDF [21], further improved sparse-data reconstruction. However, implicit methods typically require dense per-point evaluations at inference, incurring high computational costs compared to methods requiring only a single forward pass for evaluation. The point-wise querying also complicates integration into end-to-end differentiable pipelines.

### 2.4. Discretization and Resolution-Agnosticism

FNO architectures inherently provide discretization invariance since parameters are learned directly in the Fourier domain. Following Li et al. [15], our method discretizes the spatial domain uniformly, facilitating efficient Fourier transforms (via FFT). The network learns parameters independent of input resolution, enabling one-shot super-resolution by applying learned operators at higher resolutions than those used during training.

## 3. Method

In this section, we detail our Neural Poisson Surface Reconstruction (nPSR) pipeline and each component to explain how our method reconstructs watertight meshes from oriented point clouds.

Our full reconstruction pipeline comprises three main components: a point-cloud rasterization module, a Fourier Neural Operator architecture (see Fig. 2), and a threshold-based mesh extraction step. A visual overview of our pipeline is provided in Fig. 3.

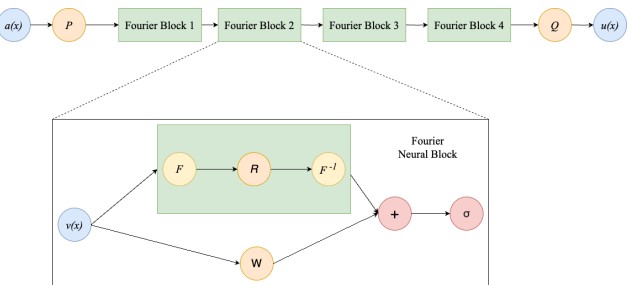

Figure 2. Fourier Neural Operator with four Fourier Blocks, the core architecture of nPSR, where $\mathcal{F}$ denotes the Fourier transform.

## 3.1. Point rasterization

The Fourier Neural Operator requires input defined on uniform grids. To transform the unstructured point cloud data into this structured representation, we rasterize the point cloud into a voxelized divergence field $\nabla \cdot V$ used in the Poisson equation (2.1). The rasterization process is composed of the following steps:

- **Step 1: Voxelization of points and normals.** We rasterize the oriented point cloud onto a regular voxel grid of resolution $n \times n \times n$ by assigning each point and its associated normal vector to the corresponding voxel, aggregating normals when multiple points fall into the same cell. The resulting discrete normal field $V \in \mathbb{R}^{n \times n \times n \times 3}$ is then smoothed with a Gaussian filter with standard deviation $\sigma = 2$ to reduce discretization artifacts.
- **Step 2: Divergence computation.** We compute the divergence $\nabla \cdot V$ using finite differences on the voxelized normal field, followed by another Gaussian smoothing step ($\sigma = 2$). This yields a smooth divergence field, approximating the right-hand side of the Poisson equation, crucial for robust surface reconstruction.
- **Step 3: Noise addition.** Finally, we add Gaussian noise (5% amplitude relative to the mean voxel values) to simulate realistic acquisition noise and enhance the robustness of the learned reconstruction.

This rasterization approach ensures the input to our neural architecture accurately reflects the underlying PDE formulation, aiding in the network's ability to implicitly solve the Poisson reconstruction task.

## 3.2. Neural operator

We use an FNO composed of four 3D Fourier blocks to solve the Poisson equation. Solving the Poisson equation amounts to convolution with its Green's function [20], which corresponds to pointwise multiplication in the Fourier domain. Since FNO layers learn multiplicative operators directly in the frequency domain, the architecture is naturally aligned with the mathematical structure of the problem. Each Fourier block is followed by GELU non-linear activations [7], with dropout layers after the first three blocks to regularize training. We retain the first $k_{\max} = 20$ frequency modes in each Fourier transform, chosen empirically to balance expressiveness and computational efficiency (see Sec. 4.4). The model is not explicitly conditioned to obey any boundary conditions.

## 3.3. Prediction thresholding and mesh extraction

The network predicts occupancy values across the voxel grid, which we subsequently threshold using Otsu's method [17] to identify occupied regions corresponding to the object's surface. A surface mesh is then extracted using the classical marching cubes algorithm [16]. The thresholding method is selected based on experimental validation (see detailed ablation studies in Sec. 4.4 and **??**).

## 3.4. Dataset preparation and sampling pipeline

We train nPSR in a supervised manner using the ShapeNet dataset [4]. We use the provided voxelized meshes as ground truth occupancy grids for the optimization of the FNO layers. The inputs are synthetically generated oriented point clouds derived from these grids using the following steps:

- **Step 1:** Mesh extraction via marching cubes [16] on voxelized shapes.
- **Step 2:** Surface sampling using Poisson disk sampling [28].
- **Step 3:** Normal vector estimation for sampled points by averaging neighboring face normals weighted by barycentric coordinates.

## 3.5. Sampling and resolution scenarios

We evaluate performance under four sampling rates: $3\,000$, $10\,000$, $25\,000$, and $250\,000$ points. Training and evaluations are performed at resolution $64 \times 64 \times 64$, with additional experiments demonstrating resolution-agnostic properties by evaluating trained models at resolutions $32 \times 32 \times 32$ and $128 \times 128 \times 128$.

## 3.6. Training and evaluation setup

We train nPSR on a representative subset of $10\,000$ shapes from ShapeNet, minimizing a mean squared error (MSE) loss between the predicted and ground-truth occupancy voxel grids. We optimize using the Adam optimizer [12] with learning rate $2.5 \times 10^{-3}$ and weight decay $10^{-4}$. Training occurs over $10\,000$ steps with a batch size of 20, feasible on consumer GPUs (Nvidia Titan RTX).

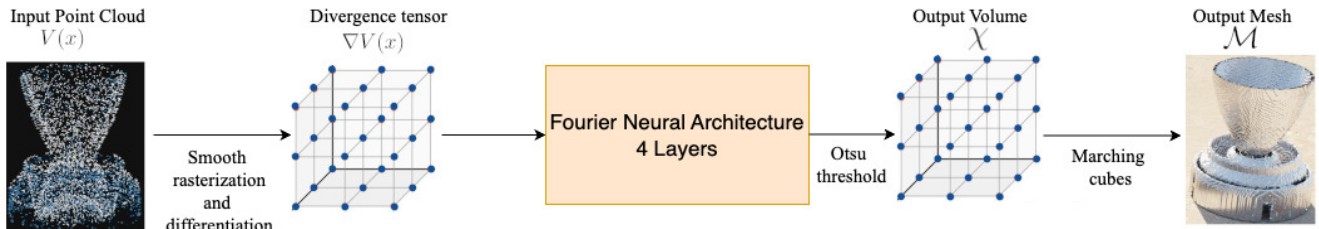

Figure 3. Full reconstruction pipeline of nPSR illustrating each stage from point cloud rasterization to mesh extraction.

| Resolution (voxels) | Chamfer-L1 ($\downarrow$) | F-Score ($\uparrow$) | Norm. Consist. ($\uparrow$) |
|---|---|---|---|
| $32 \times 32 \times 32$ | 0.024 | 0.992 | 0.983 |
| $64 \times 64 \times 64$ | 0.027 | 0.986 | 0.980 |
| $128 \times 128 \times 128$ | 0.029 | 0.983 | 0.976 |

Table 1. Reconstruction metrics averaged over all ShapeNet voxel models [4], with 25 000 sampled points. Comparison of reconstruction performance along different resolutions

Evaluation metrics include Chamfer-L1 loss, F-Score, and Normal Consistency, following established conventions [20]. We evaluate on 2 000 shapes each from ShapeNet, Thingi10K [29], and ModelNet10 [27], prepared identically to test generalization to unseen datasets.

Code and detailed instructions for reproducing all experiments are provided in the supplementary materials and will be made publicly available.

## 4. Experiments

### 4.1. Quantitative results and comparisons

We compare nPSR against classical (SPSR [10], iPSR [8] and neural-based (Point2Mesh [6], SAP [20], POCO [3], ALTO [24], and NKSR [9]) as well as neural-implicit methods (DeepSDF [18], NDC [5], GeoUDF [21]) approaches. Although methods such as iPSR and GeoUDF do not require normals, we include them to provide comprehensive benchmarks, consistent with prior studies [20].

Tab. 2 summarizes performance across different sampling rates on ShapeNet, highlighting that nPSR consistently outperforms other methods, especially in low-sampling scenarios. Supplementary tables (????) present results for ModelNet10 and Thingi10K, confirming robust generalization. Finally, ??, ??, and ?? in the supplementary materials depict the results for implicit methods on the three datasets.

nPSR demonstrates stable runtime across sampling rates, with computational costs primarily during training, ensuring straightforward scalability.

### 4.2. Resolution agnosticism

To evaluate resolution-agnostic capabilities, we train nPSR at $64^3$ resolution (with 25 000 points) and test at $32^3$ and $128^3$ resolutions. Results (Tab. 1) confirm that nPSR effectively generalizes across resolutions, achieving minimal accuracy drops even when tested at resolutions not encountered during training. Detailed results for additional datasets are provided in supplementary material (Tabs. 3 and 4). In addition we provide results comparing against implicit methods (??, ??, and ??).

### 4.3. Qualitative examples

Fig. 5 visually demonstrates reconstructions at various sampling rates, clearly illustrating superior detail and accuracy of nPSR. Additional qualitative examples across resolutions and datasets are available in supplementary materials (Fig. 4a and ????).

### 4.4. Ablation studies

We systematically investigate design choices:
- **Divergence vs. Normal Field Input**: To validate the connection of our method to Poisson Surface Reconstruction, we compared reconstruction performance when using the divergence of the normal field (as in PSR) versus using the normal field directly as input. Results in Table 8 demonstrate a substantial improvement when using the divergence field, confirming the critical role of solving the underlying Poisson equation in our approach.
- **Normal importance**: Removing normals significantly degraded performance, underscoring their critical role.
- **Gaussian smoothing**: Optimal results with Gaussian smoothing parameter $\sigma = 2$ (see Tab. 5).

| N. Sampled Points | Method | Chamfer-L1 (↓) | F-Score (↑) | Norm. Consist. (↑) | Runtime (s) |
|---|---|---|---|---|---|
| | SPSR [10] | 0.430 | 0.552 | 0.525 | 0.31 |
| | iPSR [8] | 0.554 | 0.488 | 0.465 | 325.25 |
| | Point2Mesh [6] | 0.644 | 0.392 | 0.381 | 2 215.64 |
| 3 000 | SAP [20] | 0.492 | 0.516 | 0.495 | 0.45 |
| | POCO [3] | 0.310 | 0.821 | 0.832 | 420.00 |
| | ALTO [24] | 0.300 | 0.815 | 0.835 | 0.38 |
| | NKSR [9] | 0.250 | 0.855 | 0.887 | 2.60 |
| | **nPSR** | 0.081 | 0.895 | 0.875 | 0.52 |
| | SPSR [10] | 0.265 | 0.722 | 0.715 | 0.31 |
| | iPSR [8] | 0.372 | 0.548 | 0.625 | 410.41 |
| | Point2Mesh [6] | 0.485 | 0.522 | 0.578 | 3 015.91 |
| 10 000 | SAP [20] | 0.338 | 0.852 | 0.681 | 0.45 |
| | POCO [3] | 0.130 | 0.892 | 0.860 | 420.00 |
| | ALTO [24] | 0.120 | 0.885 | 0.863 | 0.38 |
| | NKSR [9] | 0.110 | 0.900 | 0.880 | 2.60 |
| | **nPSR** | 0.042 | 0.954 | 0.912 | 0.52 |
| | SPSR [10] | 0.230 | 0.751 | 0.753 | 0.31 |
| | iPSR [8] | 0.348 | 0.583 | 0.688 | 452.33 |
| | Point2Mesh [6] | 0.412 | 0.512 | 0.623 | 3 890.82 |
| 25 000 | SAP [20] | 0.315 | 0.881 | 0.715 | 0.45 |
| | POCO [3] | 0.085 | 0.935 | 0.905 | 420.00 |
| | ALTO [24] | 0.075 | 0.930 | 0.907 | 0.38 |
| | NKSR [9] | 0.065 | 0.950 | 0.930 | 2.60 |
| | **nPSR** | 0.029 | 0.983 | 0.976 | 0.52 |
| | SPSR [10] | 0.022 | 0.954 | 0.925 | 0.31 |
| | iPSR [8] | 0.031 | 0.961 | 0.930 | 11 015.84 |
| | Point2Mesh [6] | 0.649 | 0.614 | 0.622 | 43 850.11 |
| 250 000 | SAP [20] | 0.015 | 0.974 | 0.971 | 0.45 |
| | POCO [3] | 0.012 | 0.987 | 0.982 | 420.00 |
| | ALTO [24] | 0.011 | 0.986 | 0.980 | 0.38 |
| | NKSR [9] | 0.010 | 0.988 | 0.985 | 2.60 |
| | **nPSR** | 0.008 | 0.99 | 0.991 | 0.52 |

Table 2. Reconstruction metrics averaged over all ShapeNet voxel models [4].

| Resolution (voxels) | Chamfer-L1 (↓) | F-Score (↑) | Norm. Consist. (↑) |
|---|---|---|---|
| $32 \times 32 \times 32$ | 0.021 | 0.989 | 0.979 |
| $64 \times 64 \times 64$ | 0.025 | 0.979 | 0.986 |
| $128 \times 128 \times 128$ | 0.031 | 0.983 | 0.981 |

Table 3. Reconstruction metrics averaged over all ModelNet10 voxel models [27], with 25 000 sampled points. Comparison of reconstruction performance along different resolutions

| Resolution (voxels) | Chamfer-L1 ($\downarrow$) | F-Score ($\uparrow$) | Norm. Consist. ($\uparrow$) |
| --- | --- | --- | --- |
| $32 \times 32 \times 32$ | 0.027 | 0.988 | 0.979 |
| $64 \times 64 \times 64$ | 0.031 | 0.991 | 0.979 |
| $128 \times 128 \times 128$ | 0.026 | 0.979 | 0.981 |

Table 4. Reconstruction metrics averaged over all Thingi10K voxel models [29], with 25 000 sampled points. Comparison of reconstruction performance along different resolutions

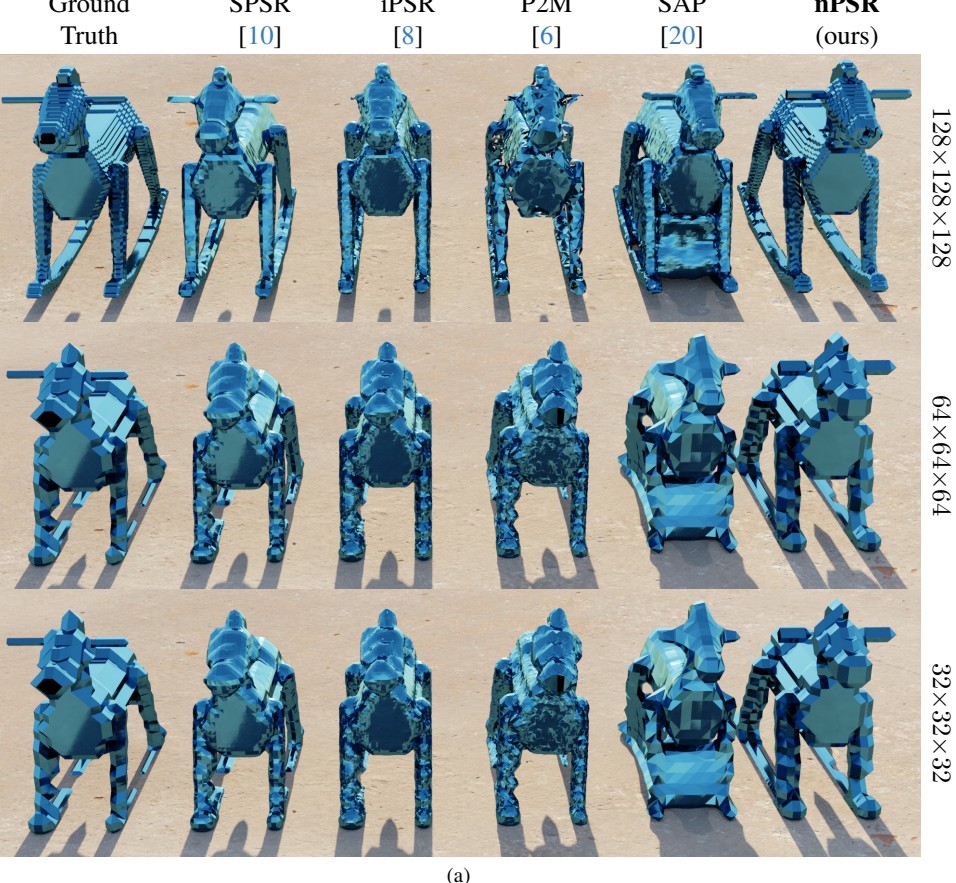

(a)

Figure 4. Our method is resolution agnostic, performing much better than existing implicit methods when increasing resolution at inference time. This figure depicts an example of the ShapeNet dataset.

- **Fourier blocks vs. convolutions**: Replacing Fourier blocks with standard 3D convolutions decreased accuracy, confirming the advantage of our Fourier-based approach (Tab. 6).
- **Thresholding**: Otsu's method [17] consistently yielded better results compared to standard iso-value thresholding (Tab. 7).

These studies validate the optimality of our pipeline choices, particularly highlighting the significance of normal data, smoothing parameters, and Fourier components.

## 4.5. Scene reconstruction

Additionally, we demonstrate that nPSR can effectively scale to real-world scene-level reconstructions. Due to GPU memory limitations, we conducted experiments at smaller scales and compared primarily against classical, non-neural baselines, as deploying neural-based methods at full-scene scales proved computationally prohibitive with our hardware. Using the SemanticKITTI dataset [1], we trained on 19k scans subdivided into $128 \times 128 \times 64$ voxel subscenes in a sliding window manner, and evaluated reconstruction quality against iPSR [8] and SPSR [11]. Our method significantly

| Standard Deviation | Chamfer-L1 ($\downarrow$) | F-Score ($\uparrow$) | Norm. Consist. ($\uparrow$) |
|---|---|---|---|
| no-smoothing | 0.378 | 0.652 | 0.520 |
| $\sigma = 0.5$ | 0.253 | 0.791 | 0.712 |
| $\sigma = 1$ | 0.112 | 0.880 | 0.810 |
| $\sigma = 2$ | 0.024 | 0.992 | 0.983 |

Table 5. Comparison of reconstruction performance for different smoothing parameters for the rasterization step. Reconstruction metrics averaged over all ShapeNet voxel models [4], with 25 000 sampled points.

| Layer type | Chamfer-L1 ($\downarrow$) | F-Score ($\uparrow$) | Norm. Consist. ($\uparrow$) |
|---|---|---|---|
| Fourier blocks | 0.024 | 0.992 | 0.983 |
| 3D convolutions | 0.250 | 0.789 | 0.764 |

Table 6. Comparison of reconstruction performance of Fourier blocks against classical 3D convolutions along different resolutions. Reconstruction metrics averaged over all ShapeNet voxel models [4], with 25 000 sampled points.

| Thresholding method | Chamfer-L1 ($\downarrow$) | F-Score ($\uparrow$) | Norm. Consist. ($\uparrow$) |
|---|---|---|---|
| Otsu [17] | 0.024 | 0.992 | 0.983 |
| Isovalue [10] | 0.175 | 0.891 | 0.764 |
| Mean | 0.248 | 0.812 | 0.720 |

Table 7. Comparison of reconstruction performance for different thresholding methods. Reconstruction metrics averaged over all ShapeNet voxel models [4], with 25 000 sampled points.

| Input Type | Chamfer-L1 ($\downarrow$) | F-Score ($\uparrow$) | Norm. Consist. ($\uparrow$) |
|---|---|---|---|
| Divergence Field (ours) | **0.024** | **0.992** | **0.983** |
| Normal Field | 0.312 | 0.645 | 0.637 |

Table 8. Ablation comparing reconstruction performance using divergence versus directly using normal fields as input. Results are averaged over ShapeNet models [4], sampled with 25 000 points.

surpassed these baselines, particularly in large-scale, sparse-data scenarios (see ?? in Supplementary and ??). However, we note that our primary focus, similar to the other presented baselines, remains object-centric reconstruction tasks.

## 5. Limitations

Despite the strong performance of nPSR, several limitations remain. The reliance on regular voxel grids imposes memory constraints that hinder scalability to large real-world scenes, making applications such as city-scale LiDAR reconstructions currently impractical. The need for oriented input normals also limits applicability when normals are missing or inaccurately estimated, affecting robustness. While nPSR is resolution-agnostic, extremely high-resolution reconstructions may lose some fine details, and occasional grid-like artifacts can reduce visual quality. Future work could ex-

plore adaptive grid structures, robust normal estimation, and improved handling of noisy inputs to enhance real-world applicability.

## 6. Conclusion

This paper presents nPSR, a novel 3D reconstruction method that achieves state-of-the-art performance, especially in low-sampling regimes ($3,000$–$25,000$ points), while also generalizing well to unseen datasets. Even at high sampling densities (up to $250,000$ points), nPSR remains competitive with leading methods.

nPSR leverages a Fourier Neural Operator backbone to efficiently solve the Poisson equation, followed by Otsu's thresholding and marching cubes for robust mesh extraction. Its one-shot super-resolution capability—training on low-resolution grids (e.g. $64^3$) yet performing well on higher

resolutions (e.g. $128^3$)—offers a key advantage over CNN-based approaches.

While currently limited to regular voxel grids, potentially restricting very high-resolution reconstructions, the resolution-agnostic design of nPSR highlights the promise of Fourier Neural Operators for future advances in shape reconstruction and related computer vision tasks.

# 7. Acknowledgments

H. Andrade-Loarca, J. Hege, and G. Kutyniok were supported by the German Research Foundation under Grant DFG-SFB/TR 109, Project C09. J. Hege and G. Kutyniok additionally acknowledge support from the Konrad Zuse School of Excellence in Reliable AI. G. Kutyniok further acknowledges support from the Munich Center for Machine Learning (BMBF). D. Cremers was supported by the ERC Advanced Grant SIMULACRON. This work is supported by the DAAD programme Konrad Zuse Schools of Excellence in Artificial Intelligence, sponsored by the Federal Ministry of Research, Technology and Space.

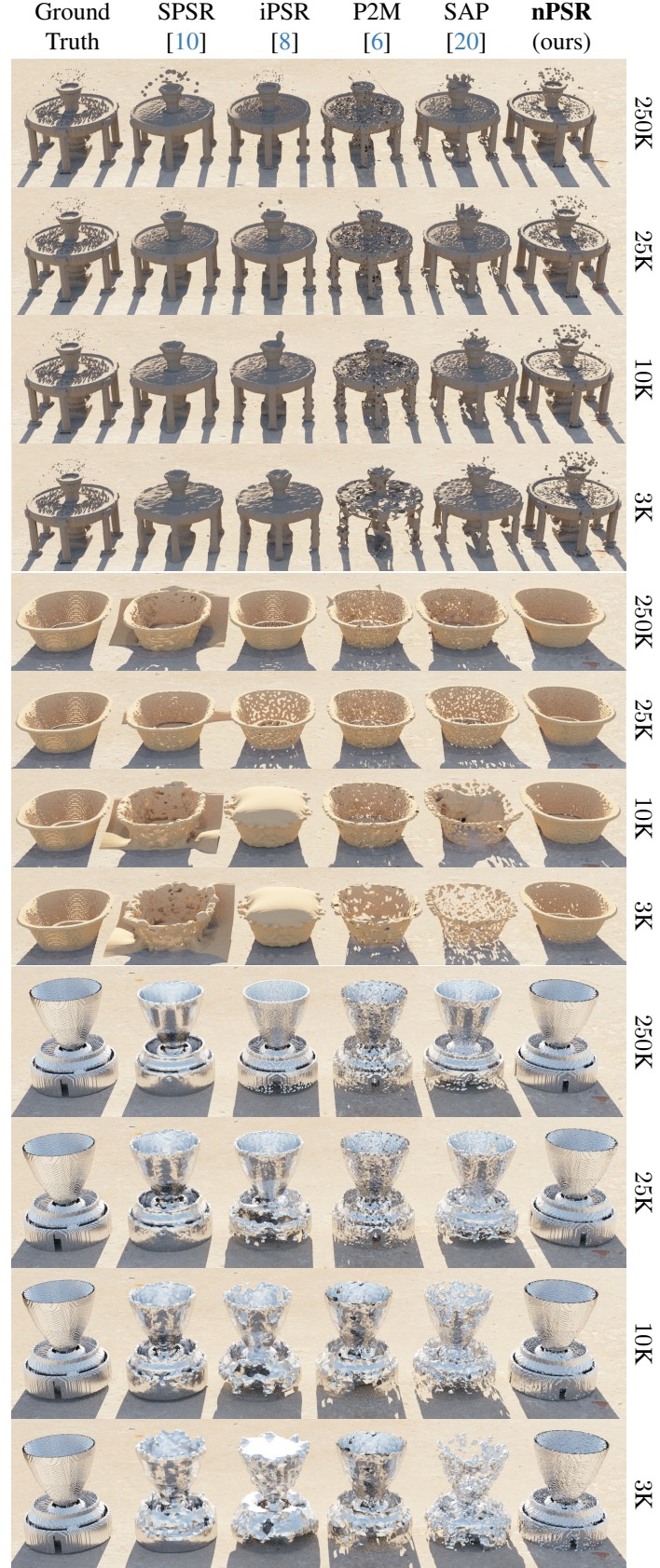

Figure 5. While all methods work well in high-sampling regimes, the proposed nPSR significantly outperforms in low-sampling regimes.

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
