# OpenReview forum: "Neural Poisson Surface Reconstruction: Resolution-Agnostic Shape Reconstruction via Fourier Neural Operators"
_thecvf.com/CVPR/2026/Workshop/3D4S — CVPR 2026 Workshop 3D4S Poster_

### Official Review · Reviewer_DBRf · 2026-04-12
**Neural Poisson Surface Reconstruction: Resolution-Agnostic Shape Reconstruction via Fourier Neural Operators**

**Rating:** 8
**Confidence:** 4

**Review:**

The paper presents a highly rigorous experimental study and demonstrates strong engineering quality, making it well-aligned with current research trends in Fourier Neural Operators (FNOs). It presents Neural Poisson Surface Reconstruction (nPSR), a hybrid framework that successfully combines classical Poisson Surface Reconstruction (PSR) with learned neural operators to tackle 3D surface reconstruction from oriented point clouds. By replacing handcrafted priors with data-driven operators, the method achieves a compelling balance between physical modeling and learning-based flexibility.

The proposed approach is both methodologically sound and practically relevant. Its design leverages the structure of the Poisson equation, $\Delta \chi(x) = \nabla \cdot V(x)$
and uses FNOs to approximate its solution efficiently in a learned manner. This enables several notable advantages, including resolution-agnostic generalization, allowing models trained on low-resolution grids to perform effectively at higher resolutions without retraining. Additionally, the method demonstrates strong robustness under sparse and noisy sampling conditions, which is critical for real-world applications.

Another key strength is its efficient single forward-pass inference, making it computationally attractive compared to iterative or query-based implicit methods. Extensive experiments across multiple datasets further validate its state-of-the-art performance. Overall, the paper satisfies major originality and methodological criteria and represents a meaningful contribution.

---

### Official Review · Reviewer_VzzK · 2026-04-13
**Outdated approximation of Poisson surface reconstruction with learned Fourier Neural Operators**

**Rating:** 5
**Confidence:** 3

**Review:**

The authors propose Neural Poisson Surface Reconstruction using Fourier Neural Operators. They formulate the solution starting with point rasterization involving voxelization and gaussian smoothing. The novelty of the paper lies in the next step of using fourier neural operators to approximate learned implicit priors and use marching cubes to extract meshes. They show that this method is scalable with resolution. However the paper is very outdated as they don't explore the more recent representations like triplane latents or structured latents. Since the final mesh is still extracted with marching cubes, the method inherits all the problems associated with it. The authors also correctly point out the memory constraints involving using voxel grids. The upside to limitations don't balance as well because the original poisson reconstruction does not require any training. I am curious to learn how the network performs on out of domain data and how much training data is required to capture the variance of a shape

---

### Decision · Program_Chairs · 2026-04-28

Accept (Poster)